# Indentation Reverse Algorithm of Mechanical Response for Elastoplastic Coatings Based on LSTM Deep Learning

**DOI:** 10.3390/ma16072617

**Published:** 2023-03-25

**Authors:** Xu Long, Xiaoyue Ding, Jiao Li, Ruipeng Dong, Yutai Su, Chao Chang

**Affiliations:** 1Research & Development Institute, Northwestern Polytechnical University in Shenzhen, Shenzhen 518063, China; 2School of Mechanics, Civil Engineering and Architecture, Northwestern Polytechnical University, Xi’an 710072, China; 3School of Applied Science, Taiyuan University of Science and Technology, Taiyuan 030024, China

**Keywords:** elastoplastic coating, indentation, reverse analysis, FE simulation, LSTM neural network

## Abstract

The load-penetration depth (*P*–*h*) curves of different metallic coating materials can be determined by nanoindentation experiments, and it is a challenge to obtain stress–strain response and elastoplastic properties directly using *P*–*h* curves. These problems can be solved by means of finite element (FE) simulation along with reverse analyses and methods, which, however, typically occupy a lengthy time, in addition to the low generality of FE methodologies for different metallic materials. To eliminate the challenges that exist in conventional FE simulations, a long short-term memory (LSTM) neural network is proposed in this study and implemented to deep learn the time series of *P*–*h* curves, which is capable of mapping *P*–*h* curves to the corresponding stress–strain responses for elastoplastic materials. Prior to the operation of the neural network, 1000 sets of indentation data of metallic coating materials were generated using the FE method as the training and validating sets. Each dataset contains a set of *P*–*h* curves as well as the corresponding stress–strain curves, which are used as input data for the network and as training targets. The proposed LSTM neural networks, with various numbers of hidden layers and hidden units, are evaluated to determine the optimal hyperparameters by comparing their loss curves. Based on the analysis of the prediction results of the network, it is concluded that the relationship between the *P*–*h* curves of metallic coating materials and their stress–strain responses is well predicted, and this relationship basically coincides with the power-law equation. Furthermore, the deep learning method based on LSTM is advantageous to interpret the elastoplastic behaviors of coating materials from indentation measurement, making the predictions of stress–strain responses much more efficient than FE analysis. The established LSTM neural network exhibits the prediction accuracy up to 97%, which is proved to reliably satisfy the engineering requirements in practice.

## 1. Introduction

Coatings are frequently employed to protect materials and structures in harsh environments and to reduce or stop damage caused by outside forces. Particularly, metallic coatings have a wide variety of applications due to their excellent properties. For instance, in the aerospace industry, titanium-based coatings are frequently used to protect aerospace engines, because of their excellent corrosion resistance and stable strength [1,2,3,4]. To achieve corrosion resistance of steel components in automotive industry, aluminum, zinc, and zinc-boron compounds are the most popular metals utilized as coatings [5].

In order to assure the coating’s reliability during applications, it is essential to understand and quantify its mechanical qualities, especially the in situ properties after the coating process with high temperature, plastic formation, and strain rate. Apparently, the coating is essentially adhered to the substrate material, and the presence of the substrate material directly affects the findings of macroscopic testing of coating materials if the substrate effect cannot be reasonably excluded. In this regard, experimental nanoindentation methods have been presented for analyzing the microscale characterization features of materials [6,7,8,9]. In instrumented nanoindentations, the indenter penetrates the material’s surface at a controlled speed or load, and the applied load *P* and the penetration depth *h* of the indenter are simultaneously recorded. By examining the *P*–*h* curves under various working situations, a deep understanding can be made for the material’s mechanical properties. By following the pioneering Oliver–Pharr model, the hardness and elastic modulus can be obtained based on the *P*–*h* curve [10,11,12,13]. Nevertheless, the material’s stress–strain response cannot be derived directly from the *P*–*h* curve. In spite of the fact that the mapping from the *P*–*h* curve to the stress–strain curve can be resolved by inverse analysis with finite element (FE) software, this is not only necessary to consider the relationship between several physical quantities during the analysis process, but also it is time-consuming to perform a parametric study of FE simulations. This means that the method simply based on exhausting FE simulations is ineffective in capturing the essence of such an inverse problem for elastoplastic materials.

In light of the aforementioned challenges in the analysis process, machine learning methods are more advantageous than FE simulation at handling nonlinear relationships of data. Machine learning techniques are utilized in industrial manufacturing due to their outstanding capabilities for data analysis [14,15]. The machine learning approach has gained popularity in material inspection analysis because of its low development cost, short development cycle, and excellent predictive performance when dealing with large amounts of data [16]. Deep learning is one of the subfields of machine learning, which has been the major research focus of computer applications in recent years. Compared with conventional machine learning, deep learning can automatically execute the majority of data feature extraction during network training and can work with significantly bigger datasets. On the other hand, machine learning concentrates on applying statistical approaches to data, but deep learning primarily imitates biological neural networks to extract data features step by step. In recent years, convolutional neural networks (CNN), recurrent neural networks (RNN), and generative adversarial networks (GAN) have been the most popular deep learning models [17,18,19,20,21]. These three types of networks have their own application scenarios based on different algorithms. Among them, CNN is frequently used to process and analyze medical images due to its powerful image analysis capability [22,23]. The special feature of GAN is that there are two networks inside it, and during the training process, the two networks fight against each other to continuously approach the optimal solution of the problem. Therefore, GAN is used to generate random data for applications such as image enhancement and data denoising [24,25,26]. LSTM is derived from RNN, compensating for the fact that RNN cells cannot effectively learn the relevant information of the input data when the input time period is lengthy [27,28,29,30,31]. LSTM is frequently used in text recognition, audio processing, and video processing due to its high effectiveness with time series data [32,33,34]. For indentation problems, Long et al. [35] adopted the *P*–*h* curves of metal materials to train the CNN network; some constitutive parameters are regarded as the training objective. Their datasets are generated by means of FE modeling, and the stress–strain relationship of the metallic material is described by a power-law equation [36,37]. However, the datasets loose certain characteristics during the convolution in the CNN network. Essentially, there is a close relationship between the *P*–*h* curve and the constitutive equation, and all data are continuous in the time history. Compared with CNN and GAN networks, a deep learning method based on LSTM could be more advantageous to optimize the reverse process of the mechanical behavior of materials in indentation studies.

In this study, an LSTM neural network is established to perform the inverse calculation of *P*–*h* curves to the stress–strain responses of metallic coating materials. Initially, 1000 datasets of matched *P*–*h* curves and stress–strain curves are constructed by performing extensive FE simulations. After extracting the coordinates of the *P*–*h* curves with equal spacing divisions, the maximum-minimum normalization technique is applied in order to generate the network’s input data. The established network then processes the input data and outputs the corresponding stress–strain response, which completes the crucial training phase for the LSTM neural network. With the trained LSTM network, some independent cases are further examined in the validating phase. This technology is promising to effectively address the challenges of conventional nanoindentation inversion work and markedly enhance computing efficiency. To guarantee the correctness of the proposed procedure, the predicted results from the LSTM network and the FE simulations are compared by in-depth discussions with various statistical evaluation indicators.

## 2. Indentation Theory and Database Preparation

### 2.1. Theoretical Basis for Instrumented Indentation

To determine the deformation properties of metallic coating materials, in nanoindentation tests, the indenter is usually controlled to penetrate into the material at a constant load/displacement speed or at a constant indentation strain rate. If no holding stage is considered, the process is separated into two stages based on the direction of the indenter’s movement: the loading and the unloading stages. By applying geometric self-similar indenters, the loading and deformation behavior of the material specimen usually satisfies Kick’s law [38,39] as given by
(1)$$P=Ch^{m},$$
where *P* is the load applied to the material by the indenter during loading, *C* represents the curvature during loading, and *h* is the depth of indentation into the surface of the material. For the exponent, *m* is equal to 2 for an ideal indention but is found to be usually less than 2 if uncertainties and imperfections of indenters and test samples are taken into account.

From the measured *P*–*h* curves, by continuously monitoring the applied load during the loading stage, material hardness can be determined and identified as a critical parameter to characterize the material properties. Hardness, defined as the average contact pressure on the contact area *A_c_* due to the applied load on the indenter, demonstrates the material’s capacity to resist local deformation, specifically plastic deformation, indentation, and scratching. The standard definition of hardness *H* is
(2)$$H=\frac{P}{A_{c}}.$$

The contact area *A_c_* can be calculated with the indentation depth for the geometrically self-similar indenter and is approximated as 24.5 *h*^2^ for a three-sided pyramid Berkovich indenter. In fact, the value of hardness was found to be closely related to the yield strength by the constraint factor (i.e., 3) as proposed by Tabor [40]. In addition to the plastic properties, the Young’s modulus, the most important elastic property, can also be determined from the *P*–*h* curves by adopting the concept of reduced modulus. Apparently, the elastoplastic properties are implied in the measured *P*–*h* curves. Even though intensive efforts [38,41,42,43,44] have been made to establish reverse algorithms in recent decades, more reliable and efficient approaches are still unavailable.

In order to identify the stress–strain variation relationship of metal-coated materials during indentation experiments, the elastoplastic properties of the material can be extracted using a reversal method based on the measured material indentation dataset, whereas for metallic or alloy materials, their plastic behavior can be approximated using a power-law constitutive model [37,45,46] in the following form of
(3)$$\sigma =\{\begin{array}{c} E\varepsilon_{}^{}\;\varepsilon \leq \varepsilon_{y}\\ R\varepsilon_{y}^{n}\;\varepsilon \geq \varepsilon_{y} \end{array},$$
where *E* is Young’s modulus, *R* is the strength factor, *n* is the strain hardening index, and *ε_y_* is the yield strain corresponding to the yield strength.

It is essential to emphasize that the purpose of this study is to propose an efficient but reliably accurate prediction method of constitutive properties for elastoplastic materials by instrumented indentations. As a powerful and effective method for data analysis, deep learning allows the dataset consisting of indentation response of materials to function as input data, whilst the output data consists of the material’s elastoplastic properties. The prediction model based on deep learning theory not only has excellent prediction capacity, but also can save a lot of time to improve the efficiency of engineering practice, which is of great help to realize the analysis of in situ mechanical properties of metallic coating materials. Therefore, the theories and formulae mentioned above are regarded as the theoretical basis for generating datasets for deep learning.

### 2.2. FE Simulation of Indentation

In order to generate sufficient reliable data, FE simulations are performed with a three-sided pyramid Berkovich. Note that for a Berkovich indenter, the hardness values of three-dimensional and axisymmetric indentations can be measured over the range from micro to nano-indentations, despite the indentation size effect. In this study, the Berkovich indenter is modeled as an axisymmetric part with a half angle of 70.3° to maintain the equivalent contact area with the three-dimensional indenter. The substrate material is a cylinder with a height of 100 μm and a radius of 100 μm, which is discretized with 49,349 axisymmetric elements. With a maximum indentation depth of 2000 nm, the substrate’s mesh size of 50 nm is sufficiently fine to capture the high strain gradient underneath the indenter. Figure 1 shows the experimental model of nano-indentation constructed during the FE simulation. It should be noted that the applied indentation depth is much smaller than the height of the substrate material. The entire indentation process is examined by fixing the radial displacement of the reference point assigned on the indenter. Post-processing work allows for accurate output of the *P*–*h* curve and stress–strain curve, laying the groundwork for reliable further numerical verification, where variations of Young’s modulus, yield strength and hardening exponent are taken into account. While employing the FE simulation approach for gathering training data, the noise in all data is uniformly reduced to improve model prediction robustness.

## 3. Methodology

### 3.1. LSTM Neural Network

With the rapid development of machine learning, deep learning as one of its primary branches has been widely applied to scientific and practical engineering problems [47], and the most appropriate deep learning algorithms can be utilized to handle practical engineering problems as effectively as feasible. In the present work, it is crucial to analyze the nanoindentation experimental data generated by the FE simulation program in order to identify an appropriate algorithm. The metallic coating materials indentation dataset has two attributes. On the one hand, it is sequential in the sense that each point on the *P*–*h* curve created by the material during the nanoindentation experiment is closely connected, which conforms to the nonlinear relationship between pressure and material deformation as stated by Kick’s law. The second factor is dimension-transformability. Each *P*–*h* curve created by the material during indentation trials is mapped by the inversion algorithm onto the associated stress–strain curve.

Based on the features of the metallic coating indentation dataset and the nature of the problem to be solved, LSTM neural networks are quite effective deep learning methods. In 1997, Jürgen Schmidhuber’s complete derivation of the LSTM algorithm, which extracts attributes of a dataset over an extended time interval, made its debut [48]. The LSTM neural network uses memory cells and gating units to control the storage and deletion of data. The gating units used are the input gate, which sets the size of the amount of data fed into the network each time. The forgetting gate controls the information that is ignored by the memory cells. The output gate determines the final output sequence feature value. The neural network’s operation at time *t* is illustrated in Figure 2. Analysis of the LSTM neural network’s data flow operation and its mathematical model can be expressed as follows [49,50,51,52,53]:(4)$$f_{t}=S(W_{f}\cdot [h_{t-1},x_{t}]+b_{t}),$$
(5)$$i_{t}=S(W_{i}\cdot [h_{t-1},x]+b_{i}),$$
(6)$${\vec{C}}_{t}=\tanh(W_{c}\cdot [h_{t-1},x]+b_{c}),$$
(7)$$C_{t}=f_{t}\times C_{t-1}+i_{t}\times \vec{C_{t}},$$
(8)$$o_{t}=S(W_{o}\cdot [h_{t-1},x_{t}]+b_{o}),$$
(9)$$h_{t}=o_{t}\times \tanh(C_{t}),$$
where *i_t_*, *f_t_*, and *o_t_* represent the input gate forgetting gate and the output gate’s result, respectively. *C_t_* represents the cell’s state at time *t*, which can be utilized to form the connection between memory cells via ${\vec{C}}_{t}$. *h_t_* represents the state of the hidden layer at moment *t*. *W_x_* and *b_x_* (*x = i*, *f*, *o*, *c*) are the weight and bias values of the matrices in the LSTM neural network for input gates, forgetting gates, output gates, and cell state updates. S (Sigmoid) and tanh indicate the activation functions utilized by the appropriate gating units.

The indentation data processed in this study contain metallic materials with a wide range of Young’s modulus, as reflected in the series over time. For different materials, the time series are different. Regarding this kind of time series, the LSTM method based on RNN can be a good solution to the problem of time series variation. The data in different time stages are understood in the same network, which can well reflect the stress–strain response pattern in each time stage of the indentation process. The key point of this model is the use of recursive network approach to understand the temporal nature of the indentation process through multi-layer calculations.

### 3.2. Normalization for Data Preparation

After producing sufficient data on metallic coating indentation by the FE model, 1000 sets of data were chosen for network training, of which 90% are training data and 10% are validation data. For data preparation, the horizontal coordinates of the *P*–*h* curve of the metal coating material during the loading stage are divided into forty points spaced by fifty nanometers, and the vertical coordinates of these forty points serve as the input for each deep learning training. In order to increase the generalization of the training network to the dataset and the stable robustness of the prediction effect, the network parameter dropout is also set during the construction of the LSTM neural network. This aims to counteract the effect of overfitting in the deep learning process and improve the network’s ability to process the data. The workflow of the LSTM neural network constructed in this paper is shown in Figure 3.

Normalization of the dataset is an essential component as well. This is due to the fact that the input data and network output data have different scales and units during the training process, as well as a considerable variation in order of magnitude, which may result in smaller feature quantities being disregarded during training. However, data normalization can not only minimize the influence of scale between input and output responses, but also make the data metrics comparable by standardizing their magnitudes. The dataset normalizing method utilized in this paper is normalization, which has the benefits of effectively enhancing the convergence speed and accuracy of the network and reducing the negative impacts caused by odd sample data. Following are the formulae for the max-min normalization approach:(10)$$X=\frac{x-x_{\min}}{x_{\max}-x_{\min}},$$
where *x* is the current input value, *X* is the input value after reduction to the interval [0,1]. *x_min_* and *x_max_* are the minimum and maximum values. In the final output stage, the inverse normalization method can be used to obtain the corresponding prediction results.

### 3.3. Evaluation Indicators

Through a great number of repetitive calculations, the training stage of the LSTM neural network consists of seeking the optimal solution to the present system of nonlinear equations in the real number range. To effectively handle this optimization problem, the adaptive moment estimation (ADAM) was selected to obtain the optimal real number solution. The ADAM optimizer enables faster gradient reduction than conventional optimization methods, resulting in more accurate updating of the weight and deviation values obtained during computation [54].

In this research, the mean square error (*MSE*) is used to calculate the loss function in the network, which indicates the error between the predicted and actual values of the network throughout each cycle, and the root mean square error (*RMSE*) is utilized to complement the error computation. In addition, for a more straightforward evaluation of the performance of the LSTM neural network, the coefficient of determination *R*^2^ is utilized to describe the correlation between the network-predicted values and the target values. The aforementioned evaluation metrics are defined by
(11)$$MSE=\frac{1}{n}\sum_{i=1}^{n}(x_{\mathrm{pred}}-x_{\mathrm{true}}{)}^{2},$$
(12)$$RMAS=\sqrt{MSE}=\sqrt{\frac{1}{n}\sum_{i=1}^{n}{\left(x_{pred}-x_{true}\right)}^{2}},$$
(13)$$R^{2}=1-\frac{\sum_{i=1}^{n}{\left(x_{pred}-x_{true}\right)}^{2}}{\sum_{i=1}^{n}{\left(x_{true}-\bar{x}\right)}^{2}}$$
where *n* represents the total number of samples, *x_pred_* represents the value predicted by the LSTM neural network, *x_true_* represents the value obtained through validation of the power-law equation, and $\bar{x}$ represents the corresponding mean value.

## 4. Results and Discussion

### 4.1. Hyperparameter Setting of LSTM Neural Network

Prior to the network training, the first step is to set the network’s hyperparameters. The adequacy of the hyperparameter settings has a direct impact on the predictive power of the final network. Consequently, the intention of the deep learning methodology is to identify the optimal numerical solution to the nonlinear system of equations under various hyperparameter settings. The ideal combination of hyperparameters for the current problem is determined by comparing the network prediction results under each set of hyperparameter settings. The most important hyperparameter is the learning rate, which indicates the decay efficiency of the network’s gradient in each cycle, which is typically set between 10^−2^ and 10^−6^. When the learning rate is increased, the gradient reduces more quickly and the network’s learning efficiency is accelerated, but the best solution will be overlooked during network calculation. Smaller learning rates are appropriate for more precise solutions, but excessively small learning rates result in prolonged training times and poor gradient reduction. Based on the relationship between the *P*–*h* curve during the nanoindentation test and the constitutive equation, the final learning rate for this study is determined as 10^−4^. The number of training cycles of the network has the same features as the learning rate, and 1000 training cycles are used to make the learning curve of the LSTM neural network smooth and stable at a low order of magnitude.

Various types of deep learning networks have unique hyperparameters based on their frameworks. For LSTM neural network, the specific hyperparameters include input size, hidden size, and number of layers, which are defined as explained by the documentation of PyTorch. The number of hidden size reflects the amount of network features in each hidden layer, whereas the number of layers specifies the number of hidden layers utilized for each operation. For the hidden size and layer number, the control variables method was used to set multiple combinations of hyperparameters, and the optimal combination was determined by comparing the loss curves generated from each group. The number of hidden layers is set from 1 to 3, and the hidden size is set to 20, 40, 60, and 80 for each hidden layer, respectively. Upon completion of the test, the loss curves for various combinations are depicted in Figure 4 and Figure 5. The solid lines in Figure 4 and Figure 5 show the loss curves for the training stage, while the dashed lines indicate the loss curves for the validation stage.

Once the loss function’s value is absolutely minimal, it indicates that the network’s predicted value exactly approximates the actual value. Focusing on the loss curves for the combinations of hyperparameters in Figure 4 and Figure 5, the change trend of the loss curve with different combinations of hyperparameters can be initially determined. The larger the hidden size and the number of layers, the steeper the loss curve’s decline and the lower the loss value. For the findings depicted in Figure 4 and Figure 5, the LSTM neural network with three hidden layers and 80 hidden units in each hidden layer provides the most effective training performance. During the testing of the optimal hyperparameter combinations for the 12 sets of LSTM neural networks shown above, each network took about 7000 s to complete a full calculation, while the difference value between the longest and shortest run times was only 800 s.

In order to determine the optimal combination of hyperparameters, additional tests are conducted to verify the validity of this trend. Figure 6 depicts the outcomes of the extra tests. Additional studies demonstrate that increasing the number of hidden layers or hidden units has a little influence on the loss curve, and that the final loss remains stable and there is no significant decrease compared to the loss curves in Figure 4 and Figure 5. Adding extra hidden layers or hidden units during network training will dramatically increase the computation and lengthen the training duration. When the number of hidden layers is larger than three or the number of hidden units is greater than or equal to 120, network training requires much more time. This result is approximately 30% longer than the test findings presented in Figure 4 and Figure 5. The combination of hyperparameters for each network and the corresponding run times can be found in Table A1 in Appendix A. Considering the training efficiency and network prediction accuracy, hidden size and number of layers are set to 100 and 3, respectively.

### 4.2. LSTM Neural Network Performance Evaluation

To ensure the data-processing resilience of our LSTM neural network, the data used in the training and validation sets in this paper are separated. In addition, the data used for prediction are newly created by the FE simulation when network training is complete. In this study, the coefficient of determination *R*^2^ and the root mean square error (*RMSE*) are used to evaluate the LSTM neural network. The closer the value of *R*^2^ is to 1, the smaller the difference between the prediction result of the network and the actual value, and the closer the value of *RMSE* is to zero, the more accurate the network’s prediction is. Forty sets of test data are used to put into the network for the validation of the prediction effect. Figure 7 shows the prediction results and the mean absolute percentage error (*MAPE*) distribution. The formula for the *MAPE* is defined by
(14)$$MAPE=\frac{1}{n}\sum_{i=1}^{n}\left|\frac{x_{pred}-x_{true}}{x_{true}}\right|\times 100\%,$$

Each set of test data contains 40 vertical coordinate points on the *P*–*h* curve, corresponding to the pressure values applied by the indenter during the indentation test. After the test data are calculated by the LSTM neural network, the network outputs a stress–strain curve that describes the mechanical behavior of the metal coating material.

The results in Figure 8 show that the LSTM neural network established in this paper can accurately predict the stress–strain response of metallic materials. More importantly, the variation pattern of the predicted values of the network output is consistent with the trend of the elastoplastic behavior of the metallic material described by the power-law equation. In order to observe the performance of the LSTM neural network in practice more intuitively, a portion of the sample data is randomly selected as listed in Table 1 for further demonstration of the accuracy of stress–strain response of metallic materials.

As listed in Table 1, the Materials 1 to 4 typically represent the datasets prepared in the FE simulation and LSTM network predictions. The solid line indicates the constitutive relationship of metallic material used for the FE simulation, and the dashed line indicates the stress–strain response predicted by the proposed LSTM neural network from the corresponding *P*–*h* response.

## 5. Conclusions

In this paper, an LSTM-based deep learning method for completing the inversion of *P*–*h* curves to material stress–strain response is provided. The prediction results of the network achieved satisfactory results, with a value of 0.8645 for the coefficient of determination *R*^2^. Moreover, based on the *MAPE* distribution of the test samples, the prediction accuracy of the LSTM neural network is as high as 97.11. These satisfactory evaluation indicators demonstrate that the network prediction values are generally in agreement with the FE simulation values. The present study explored a wide range of materials with Young’s modulus ranging from 200 GPa to 500 GPa. The deep learning model presented in this paper is applicable to indentation data for the vast majority of metallic materials. This study provides more evidence that the power-law instanton may accurately characterize the mechanical behavior of the vast majority of metallic materials. In other words, using a deep learning approach, the relationship between the material’s intrinsic structure and the given material indentation data can be effectively reversed, which can then be employed directly in FE computations. The LSTM neural network inverse performance of the intrinsic structure relationship is applicable to a broader range of parameters, and the training dataset includes the majority of typical metallic coating materials, including magnesium, tungsten, and their compounds. The raw data also includes elastic and plastic data, which can comprehensively characterize the mechanical behavior of metallic materials. In comparison to traditional FE modeling and nanoindentation experiments, the LSTM neural network minimizes the consumption of experimental materials on the one hand and greatly improves computational efficiency on the other.

Due to space limitations, the deep learning method proposed in this study is only used for back-calculating the mechanical behavior of materials in the field of nanoindentation research, and only a single power-law equation is used in the generation of the dataset. It is worth noting that deep learning, as a data analysis method, is applicable for arbitrary intrinsic structure models. With the availability of the corresponding dataset, the matching numerical relationships can be derived by network operations. Therefore, it is a desirable way to use deep learning methods to predict the service life of coating structures and even the development of cracks in future research.

## Figures and Tables

**Figure 1 materials-16-02617-f001:**
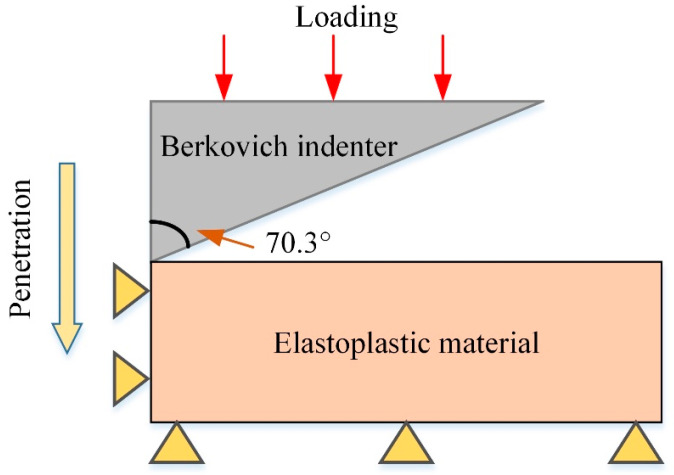
Schematic diagram of FE model for simulating the indentation by a Berkovich indenter.

**Figure 2 materials-16-02617-f002:**
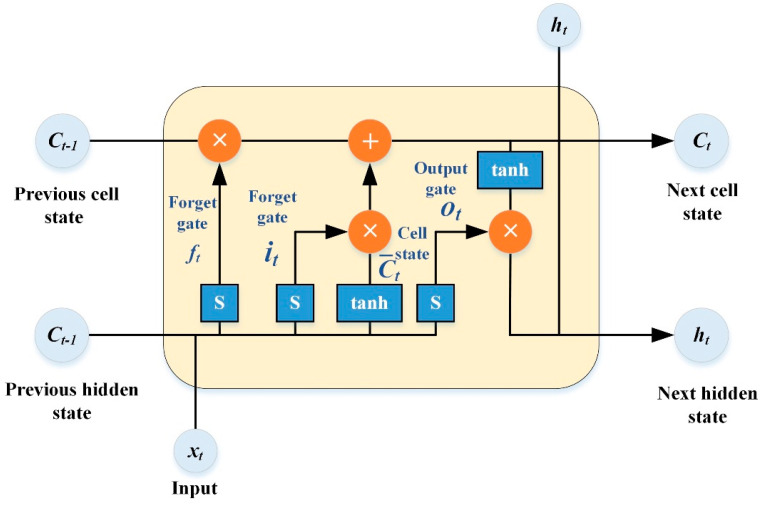
LSTM working schematic at time *t*.

**Figure 3 materials-16-02617-f003:**
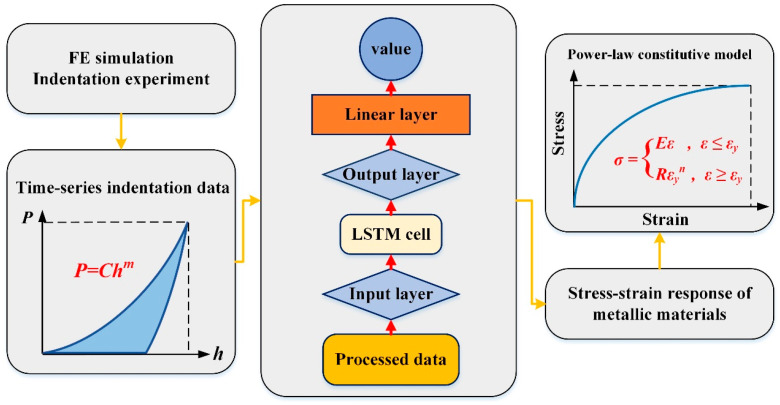
Work process of LSTM neural network.

**Figure 4 materials-16-02617-f004:**
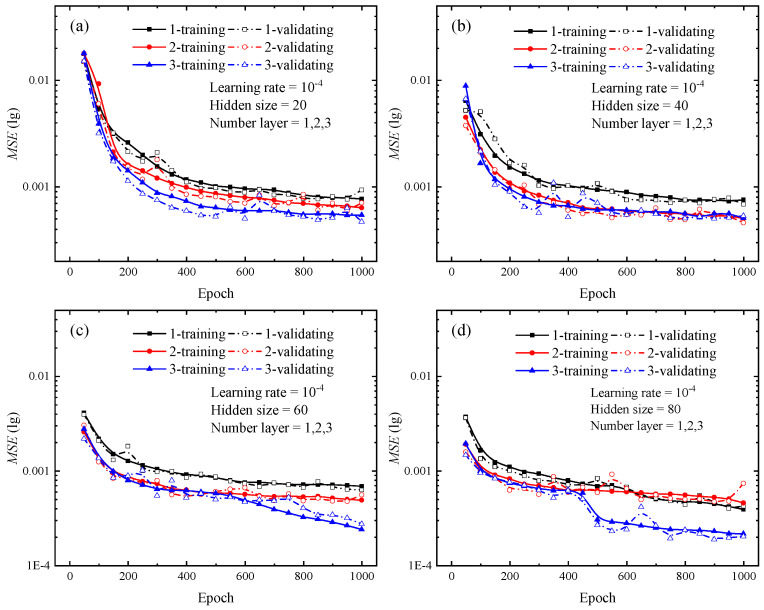
Loss curves of different numbers of hidden layers during the training and validating phrases for indentation predictions. (**a**) Hidden size of 20; (**b**) Hidden size of 40; (**c**) Hidden size of 60; (**d**) Hidden size of 80. In the legend, the cases of 1, 2, and 3 represent the number of hidden layers in the LSTM neural network.

**Figure 5 materials-16-02617-f005:**
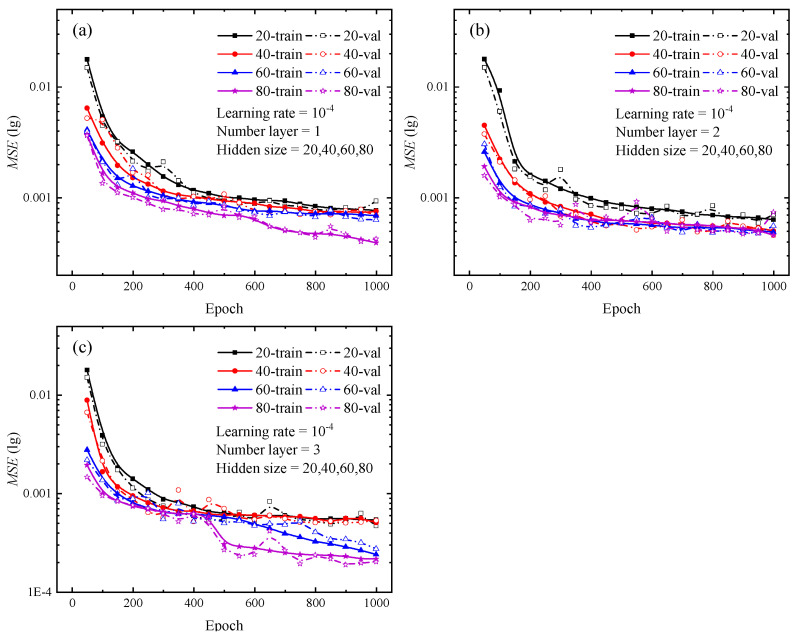
Loss curves of different hidden sizes during the training and validating phrases for indentation predictions. (**a**) The number of hidden layers is 1; (**b**) The number of hidden layers is 2; (**c**) The number of hidden layers is 3. In the legend, the cases of 20, 40, 60 and 80 represent the number of hidden sizes the LSTM neural network.

**Figure 6 materials-16-02617-f006:**
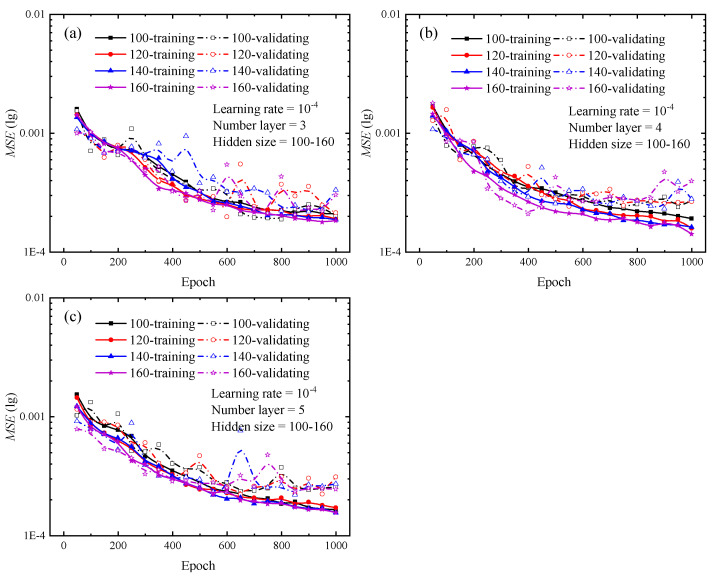
Loss curves of different hidden sizes during the training and validating phrases for indentation predictions. (**a**) The number of hidden layers is 3; (**b**) The number of hidden layers is 4; (**c**) The number of hidden layers is 5. In the legend, the cases of 100, 120, 140 and 160 represent the number of hidden sizes the LSTM neural network.

**Figure 7 materials-16-02617-f007:**
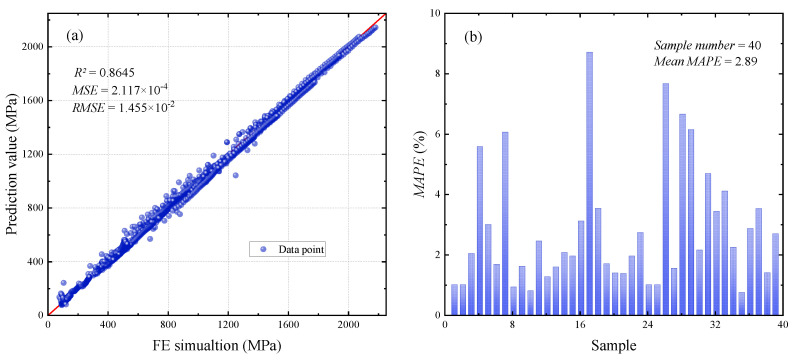
The prediction performance of the LSTM neural network is demonstrated. (**a**) indicates the prediction effect of the LSTM neural network on 40 sets of sample data, and the predicted stress values are concentrated on the diagonal line of figure. (**a**,**b**) shows the MAPE distribution of each sample.

**Figure 8 materials-16-02617-f008:**
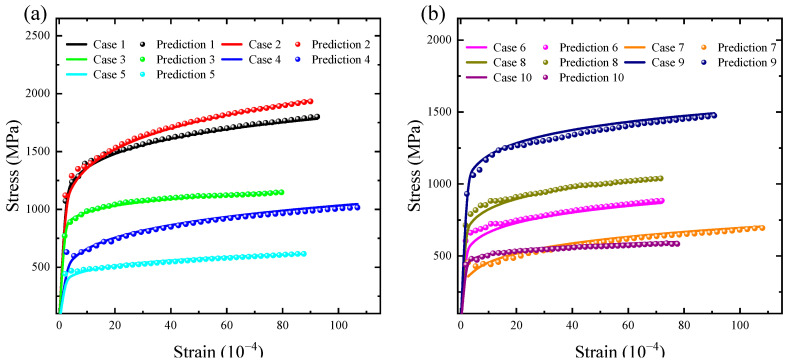
Comparison of predicted stress–strain responses by FE model and the proposed and LSTM neural network (**a**,**b**).

**Table 1 materials-16-02617-t001:** The constitutive model parameters of the 10 specimens used in the model predictions.

Parameter	*E* (GPa)	*R* (GPa)	*ɛ_y_*	*n*
Case 1	492	2.38	2.31 × 10^−3^	1.22 × 10^−1^
Case 2	473	2.87	2.25 × 10^−3^	1.63 × 10^−1^
Case 3	430	1.41	1.99 × 10^−3^	7.93 × 10^−2^
Case 4	191	1.62	2.67 × 10^−3^	1.95 × 10^−1^
Case 5	178	0.83	2.19 × 10^−3^	1.24 × 10^−1^
Case 6	293	1.24	1.80 × 10^−3^	1.36 × 10^−1^
Case 7	132	1.06	2.70 × 10^−3^	1.84 × 10^−1^
Case 8	373	1.44	1.79 × 10^−3^	1.21 × 10^−1^
Case 9	290	0.79	2.27 × 10^−3^	3.06 × 10^−2^
Case 10	224	1.16	1.94 × 10^−3^	1.57 × 10^−2^

## Data Availability

Data is unavailable due to privacy.

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
