# Peer review of "Indentation Reverse Algorithm of Mechanical Response for Elastoplastic Coatings Based on LSTM Deep Learning"

_materials, 2023, doi:10.3390/ma16072617_

Round 1

Reviewer 1 Report

This study demonstrates an intelligent approach for developing a regression model and thus supports the prognostics strategy, which is the requirement of the ongoing revolution of industry. The manuscript is articulated well and has potential practical implications. This study is also interested in the academic points of view. There are a few issues that must be addressed before my final recommendation.

Check that the style of writing is in the third person throughout. Don’t use ‘we’.

Check that the abstract provides an accurate synopsis of the paper. It is very vague in its present form.

Was the data normalized/ standardized?

How to deal with the diversity between the data distributions of present and future moments? ML-based algorithms can only resolve the regression issues within the same data distributions. What would be the key steps of generalizing to unknown moments in predicting various parameters?

Hyperparameters of each algorithm must be included in Tabular form by adding an annexure at the end of the paper.

Comment on computational time and complexity in the training of the algorithm.

Insufficient literature is presented to support the aim of the study. You may refer to the following paper to discuss the competence of deep learning algorithms. This point still needs further revision. Some reference papers are Development of Deep Belief Network for Tool Faults Recognition, Multi-Point Face Milling Tool Condition Monitoring Through Vibration Spectrogram and LSTM-Autoencoder.

How to ensure the robustness of the model in a highly noisy environment?

Whether the real-time experiment performed for collecting data? Or is it collected from simulations? It is confusing. The data collection process through experimentation, if applicable, details of data acquisition with sensors, etc., are not found anywhere and must be included. How were experiments designed? A real-time picture of the experimental setup is also missing.

Discuss the applicability of this method if there is a change in the material and its behavior.

Also, loading and unloading conditions & their effect on materials behavior are unclear and hard to understand. What about loadings such as bending, compression, shearing, or anything else? What is their effect?

I’ve suggested a few of the articles just for your reference and hope that these articles direct and guide you in your future work. If you find them worthy and interesting, you may refer them. 

All the best. Looking to receiving a revision of your manuscript.

Author Response

Response to Review 1

This study demonstrates an intelligent approach for developing a regression model and thus supports the prognostics strategy, which is the requirement of the ongoing revolution of industry. The manuscript is articulated well and has potential practical implications. This study is also interested in the academic points of view. There are a few issues that must be addressed before my final recommendation.

Response: The authors appreciate the valuable and positive comments from the reviewer. The authors’ answers to the comments are provided as follows and the manuscript has been accordingly revised and highlighted in yellow.

Check that the style of writing is in the third person throughout. Don’t use ‘we’.

Response: The authors agree with the reviewer’s points. We checked throughout the manuscript and use the style of writing in the third person.

Revisions:

In section 3.1 on page 5, “In the present work, it is crucial to analyze the nanoindentation experimental data generated by the FE simulation program in order to identify an appropriate algorithm.”

In section 3.2 on page 6, “In order to increase the generalization of the training network to the dataset and the stable robustness of the prediction effect, the network parameter dropout is also set during the construction of the LSTM neural network.”

In section 3.3 on page 7, “To effectively handle this optimization problem, the adaptive moment estimation (ADAM) be selected optimizer to obtain the optimal real number solution.”

In section 4.1 on page 8, “The ideal combination of hyperparameters for the current problem is determined by comparing the network prediction results under each set of hyperparameter settings.”

In section 4.1 on page 9, “Focusing at the loss curves for the combinations of hyperparameters in Figure 4 and 5, the change trend of the loss curve with different combinations of hyperparameters can be initially determined.”

In section 4.1 on page 10, “Considering the training efficiency and network prediction accuracy, hide size and number of layers are set to 100 and 3, respectively.”

In section 4.2 on page 11, “The results in Figure 8 show that the LSTM neural network established in this paper can accurately predict the stress–strain response of metallic materials.”

In section 5 on page 12, “In other words, using a deep learning approach, the relationship between the material's intrinsic structure and the given material indentation data can be effectively reversed, which can then be employed directly in FE computations.”

In section 5 on page 12, “With the availability of the corresponding dataset, the matching numerical relationships can be derived by network operations. Therefore, it is a desirable way to use deep learning methods to predict the service life of coating structures and even the development of cracks in future researches.”

Check that the abstract provides an accurate synopsis of the paper. It is very vague in its present form.

Response: The authors appreciate the reviewer’s feedback. We revised the abstract to ensure that it accurately summarizes the study and highlights its contributions. The authors have provided a new abstract for reviewers' reference.

Revisions: “The load-penetration depth (P–h) curves of different metallic coating materials can be determined by nanoindentation experiments, and it is a challenge to obtain stress–strain response and elastoplastic properties directly using P–h curves. These problems can be solved by means of finite element (FE) simulation along with reverse analyses and methods which, however, typically occupy a lengthy time, in addition to the low generality of FE methodologies for different metallic materials. To eliminate the challenges that exist in conventional FE simulations, a long short-term memory (LSTM) neural network is proposed in this study and implemented to deeply learn the time series of P–h curves, which is capable of mapping P–h curves to the corresponding stress–strain responses for elastoplastic materials. Prior to the operation of the neural network, 1000 sets of indentation data of metallic coating materials were generated using the FE method as the training and validating sets. Each dataset contains a set of P–h curves as well as the corresponding stress–strain curves, which are used as input data for the network and as training targets. The proposed LSTM neural networks with various numbers of hidden layers and hidden units are evaluated to determine the optimal hyperparameters by comparing their loss curves. Based on the analysis of the prediction results of the network, it is concluded that the relationship between the P–h curves of metallic coating materials and their stress–strain responses is well predicted, and this relationship basically coincides with the power-law equation. Furthermore, the deep learning method based on LSTM is advantageous to interpret the elastoplastic behaviors of coating materials from indentation measurement, making the predictions of stress–strain responses much more efficient than FE analysis. The established LSTM neural network exhibits the prediction accuracy up to 97%, which is proved to reliably satisfy the engineering requirements in practice.”

Was the data normalized/ standardized?

Response: The authors normalized/normalized the data in the current study. Based on the feedback given by the reviewers, we highlighted the data normalization processing in Section 3.2.

Revisions: “Normalization of the dataset is an essential component as well. This is due to the fact that the input data and network output data have different scales and units during the training process, as well as a considerable variation in order of magnitude, which may result in smaller feature quantities being disregarded during training. However, data normalization can not only minimize the influence of scale between input and output responses, but also make the data metrics comparable by standardizing their magnitudes. The dataset normalizing method utilized in this paper is normalization, which has the benefits of effectively enhancing the convergence speed and accuracy of the network and reducing the negative impacts caused by odd sample data. Following are the formulae for the max-min normalization approach

,

where x is the current input value, X is the input value after reduction to the interval [0,1]. xmin and xmax are the minimum and maximum values. In the final output stage, the inverse normalization method can be used to obtain the corresponding prediction results.”

How to deal with the diversity between the data distributions of present and future moments? ML-based algorithms can only resolve the regression issues within the same data distributions. What would be the key steps of generalizing to unknown moments in predicting various parameters?

Response: Thank the reviewers for their suggestions. The indentation data we deal with is diverse, as reflected in a time series that changes over time. The time series is diverse for different materials. We face this kind of time series, we use the RNN-based method, which can well solve the problem of time series variation, that is, the data of different time stages have been understood in the same network, which can well reflect the stress-strain response law of each time stage of the indentation process. The key point of this model is the use of a recurrent network approach, where multiple layers are recycled to understand the temporal indentation process.

Revisions: The authors have added new content in section 3.1 on page 5. “The indentation data processed in this study contain metallic materials with a wide range of Young's modulus, as reflected in the series over time. For different materials, the time series are different. Regarding this kind of time series, the LSTM method based on RNN can be a good solution to the problem of time series variation. The data in different time stages are understood in the same network, which can well reflect the stress–strain response pattern in each time stage of the indentation process. The key point of this model is the use of recursive network approach to understand the temporal nature of the indentation process through multi-layer calculations.”

Hyperparameters of each algorithm must be included in Tabular form by adding an annexure at the end of the paper.

Response & Revisions: Thanks to the suggestions provided by the reviewers, we have added Appendix 1 at the end of the paper, showing the combination of hyperparameters for each algorithm.

Comment on computational time and complexity in the training of the algorithm.

Response & Revisions: The authors appreciate the reviewers' inquiries about this issue. The training time for each algorithm is added to Appendix 1. For the complexity of the algorithm, the larger the value of hidden layer as well as hidden size, the greater the complexity of the algorithm.

Insufficient literature is presented to support the aim of the study. You may refer to the following paper to discuss the competence of deep learning algorithms. This point still needs further revision. Some reference papers are Development of Deep Belief Network for Tool Faults Recognition, Multi-Point Face Milling Tool Condition Monitoring Through Vibration Spectrogram and LSTM-Autoencoder.

Response & Revisions: The authors thank the reviewers for the references provided and have added relevant content in the section 1. The authors have added the following to Section 1 based on suggestions provided by the reviewers. “In recent years, convolutional neural networks (CNN), recurrent neural networks (RNN), and generative adversarial networks (GAN) have been the most popular deep learning models [17-21]. These three types of networks have their own application scenarios based on different algorithms. Among them, CNN is frequently used to process and analyze medical images due to its powerful image analysis capability [22, 23]. The special feature of GAN is that there are two networks inside it, and during the training process, the two networks fight against each other to continuously approach the optimal solution of the problem. Therefore, GAN is used to generate random data for applications such as image enhancement and data denoising [24-26]. LSTM is derived from RNN, compensating for the fact that RNN cells cannot effectively learn the relevant information of the input data when the input time period is lengthy [27-31]. LSTM is frequently used in text recognition, audio processing, and video processing due to its high effectiveness with time-series data [32-34].”

How to ensure the robustness of the model in a highly noisy environment?

Response & Revisions: Thanks to the reviewer for the question. Currently we are targeting simulation data, for high noise environment, we can choose noise reduction process before prediction. The authors have added a new description in section 2.2 on page 4 of the manuscript. “While employing the FE simulation approach for gathering training data, the noise in all data is uniformly reduced to improve model prediction robustness.”

Whether the real-time experiment performed for collecting data? Or is it collected from simulations? It is confusing. The data collection process through experimentation, if applicable, details of data acquisition with sensors, etc., are not found anywhere and must be included. How were experiments designed? A real-time picture of the experimental setup is also missing.

Response: The data used in this study were generated by the finite element simulation method. The ground truth data were obtained from a large number of numerical experiments, as the variation of experimental data affects the depth of indentation, maximum stress and strain. This part is mentioned by the authors in section 2.2 of the paper. “In order to generate sufficient reliable data, FE simulations are performed with a three-sided pyramid Berkovich.”

Discuss the applicability of this method if there is a change in the material and its behavior.

Response: We thank the reviewers for their suggestions on this paper. The present study explored a wide range of materials with Young's modulus ranging from 200 GPa to 500 GPa. The deep learning model presented in this paper is applicable to indentation data for the vast majority of metallic materials.

Also, loading and unloading conditions & their effect on materials behavior are unclear and hard to understand. What about loadings such as bending, compression, shearing, or anything else? What is their effect?

Response: In this study, we focus on the indentation experiments, and the data used in the training as well as the prediction process correspond to the loading phase of the indenter.

I’ve suggested a few of the articles just for your reference and hope that these articles direct and guide you in your future work. If you find them worthy and interesting, you may refer them. All the best. Looking to receiving a revision of your manuscript.

Response: The authors are very grateful to the reviewers and cited the recommended references.

Reviewer 2 Report

The paper analyses the topic of LSTM neural 25 networks with various numbers of hidden layers and hidden units to determine the 26 optimal hyperparameters by comparing their loss curves. The paper is well written and documented. The topic of neuronal network is original, and few treated into the literature especially considering specific applications. As it is new, I suggest to describe it better. Two parts lack of information. First, in the introduction you should describe its possibile applications, highlighting the methodologies used in different disciplines. These references for two literature review could be useful https://journalofbigdata.springeropen.com/articles/10.1186/s40537-021-00444-8 on general techniques for deep learning and https://doi.org/10.1016/j.enbuild.2022.112029 on possibile application of neuronal networks. Describe also the difference among neural networks (CNN), recurrent neural networks (RNN), and generative adversarial networks (GAN). Explain also better the gap into the literature and the novelty of your study. This aspect should be treated also in the conclusions, to illustrate further possibilities of your study. Second, the selection of the validation procedure. Why do you select these parameters? Are they the best choice? Are there specific studies on the selection of the validation parameters? Finally, conclusions lack of information, such as key findings, novelty, innovative aspect of your research, limitations, and further development. The method is clear and well addressed.

Author Response

Response to Review 2

The paper analyses the topic of LSTM neural 25 networks with various numbers of hidden layers and hidden units to determine the 26 optimal hyperparameters by comparing their loss curves. The paper is well written and documented. The topic of neuronal network is original, and few treated into the literature especially considering specific applications. As it is new, I suggest to describe it better. Two parts lack of information.

First, in the introduction you should describe its possibile applications, highlighting the methodologies used in different disciplines. These references for two literature review could be useful https://journalofbigdata.springeropen.com/articles/10.1186/s40537-021-00444-8 on general techniques for deep learning and https://doi.org/10.1016/j.enbuild.2022.112029 on possibile application of neuronal networks. Describe also the difference among neural networks (CNN), recurrent neural networks (RNN), and generative adversarial networks (GAN). Explain also better the gap into the literature and the novelty of your study. This aspect should be treated also in the conclusions, to illustrate further possibilities of your study.

Response & Revisions: The authors would like to express their sincere gratitude to the reviewers for their comments. We have adopted the suggestions made by the reviewers and firstly, which improved the introduction section. The differences between CNN, RNN and GAN are added in the introduction. In Section 1, the authors have added new references and used the relevant articles provided by the reviewers, with the following revisions: “In recent years, convolutional neural networks (CNN), recurrent neural networks (RNN), and generative adversarial networks (GAN) have been the most popular deep learning models [17-21]. These three types of networks have their own application scenarios based on different algorithms. Among them, CNN is frequently used to process and analyze medical images due to its powerful image analysis capability [22, 23]. The special feature of GAN is that there are two networks inside it, and during the training process, the two networks fight against each other to continuously approach the optimal solution of the problem. Therefore, GAN is used to generate random data for applications such as image enhancement and data denoising [24-26]. LSTM is derived from RNN, compensating for the fact that RNN cells cannot effectively learn the relevant information of the input data when the input time period is lengthy [27-31]. LSTM is frequently used in text recognition, audio processing, and video processing due to its high effectiveness with time-series data [32-34].”

Second, the selection of the validation procedure. Why do you select these parameters? Are they the best choice? Are there specific studies on the selection of the validation parameters?

Response & Revisions: The author have added Appendix 1 to show the operation time of the model under each hyperparameter combination. The hyperparameter combinations are determined by combining the decline of the loss curve and the computation time of the corresponding model. The number of hidden layers is 3 and the hidden size is 100, the model runs in a shorter time and the loss curve decreases to a lower level. In addition, we also compare the prediction results of the model with different combinations of hyperparameters and finally decide the best combination of hyperparameters.

Finally, conclusions lack of information, such as key findings, novelty, innovative aspect of your research, limitations, and further development. The method is clear and well addressed.

Response & Revisions: The author improve and supplement the conclusions of the paper so as to better reflect the innovative points of this paper. The authors have further refined the conclusions of the article as follows: “The indentation data processed in this study contain metallic materials with a wide range of Young's modulus, as reflected in the time series over time. For different materials, the time series are different. Facing this kind of time series, the LSTM method based on RNN can be a good solution to the problem of time series variation. The data in different time stages are understood in the same network, which can well reflect the stress-strain response pattern in each time stage of the indentation process. The key point of this model is the use of recursive network approach to understand the temporal nature of the indentation process through multi-layer calculation. These satisfactory evaluation indicators demonstrate that the network prediction values are generally in agreement with the FE simulation values. The present study explored a wide range of materials with Young's modulus ranging from 200 GPa to 500 GPa.”

Reviewer 3 Report

The manuscript "Indentation reverse algorithm of mechanical response for elastoplastic coatings based on LSTM deep learning" has been reviewed.

A deep learning method is proposed for back-calculating the mechanical behavior of materials in the field of nanoindentation research. The prediction results of the network achieved satisfactory results. The prediction accuracy of the LSTM neural network is declared 97.11.

In comparison to traditional FE modeling and nanoindentation experiments, the LSTM neural network minimizes the consumption of experimental materials on the one hand and greatly improves computational efficiency on the other.

In my opinion the manuscript is novel and relatively interesting, well organized with pictures and tables. Also references are almost recent and relevant for the arguments. It is clear and well argumented in results, discussion and conclusions. English, despite not mother tongue, is acceptable.

I don't have any particular concerns or remarks and, in my opinion, the manuscript can be accepted as it is.

Author Response

Response to Review 3

The manuscript "Indentation reverse algorithm of mechanical response for elastoplastic coatings based on LSTM deep learning" has been reviewed.

A deep learning method is proposed for back-calculating the mechanical behavior of materials in the field of nanoindentation research. The prediction results of the network achieved satisfactory results. The prediction accuracy of the LSTM neural network is declared 97.11.

In comparison to traditional FE modeling and nanoindentation experiments, the LSTM neural network minimizes the consumption of experimental materials on the one hand and greatly improves computational efficiency on the other.

In my opinion the manuscript is novel and relatively interesting, well organized with pictures and tables. Also references are almost recent and relevant for the arguments. It is clear and well argumented in results, discussion and conclusions. English, despite not mother tongue, is acceptable.

I don't have any particular concerns or remarks and, in my opinion, the manuscript can be accepted as it is.

Response: The support of the reviewers for this study is appreciated. The authors will try to improve their English in their future work to make the written article more impressive.

Round 2

Reviewer 1 Report

The authors have addressed all my comments positively. Congratulations. I am recommending acceptance.